# A QUALITATIVE THEORY OF DYNAMICAL SYSTEMS FOR ASSESSING STABILITY IN RESNETS

## ABSTRACT

We present an experimental method for evaluating the stability of ResNets, inspired by the qualitative theory of dynamical systems. To apply qualitative and quantitative properties from the literature on dynamical systems, we have proposed ResNets designed to maintain dimensionality throughout the residual blocks. As a result, we can not only introduce a well-suited concept of expansivity and shadowing properties for ResNets but also analyze their numerical degrees based on dynamical systems theory. This work aims to contribute to the understanding of ResNets' stability and bridge the gap between theory and practical applications.

## 1 INTRODUCTION

Deep residual networks (ResNets) He et al. (2016) began by highlighting successful tasks that have been studied with dynamical systems induced by ODEs/PDEs. This connection was first established by E (2017); Haber & Ruthotto (2017); Chang et al. (2017), and it has developed for various perspectives, such as approximation Li et al. (2022); Jung & Rojas (2023); Zhang et al. (2020), optimization/generalization Ott et al. (2020); Zhang et al. (2019); Zhang & Schaeffer (2020), and the interpretability of structures Ruthotto & Haber (2020); Lu et al. (2018); Ciccone et al. (2018). However, the theory of dynamical systems itself for ResNets has not yet made significant progress.

Inspired by the concept of Deep Learning via Dynamical Systems E (2017); Li et al. (2022), we primarily consider the *Qualitative Theory of Dynamical Systems*, which has theoretical tools to scrutinize complex phenomena without explicitly finding solutions. This study aims to develop a framework for ResNets and dynamical systems based on solid theorems.

In particular, our work focuses on topological spaces that allow a top-down approach to ResNets via dynamical systems. To apply the perspective mentioned above, we directly utilize the qualitative properties of discrete dynamical systems in flow-induced function spaces for ResNets. Fundamentally, a dynamical system consists of a phase space and a predetermined *rule* governing point evolution over *time*. Regarding this paper, we investigate ResNets within metric spaces, where the rule corresponds to *the type of skip connection*, and time is represented by *the number of residual blocks*. As a result, we present qualitative properties of topological dynamics along with accompanying numerical experiments designed to interpret and evaluate ResNets.

The primary processes outlined in this work are as follows:

- First, we establish a theoretical framework inspired by dynamical systems that corresponds to experiments in supervised learning. We empirically demonstrate various properties of dynamical systems;

- Taking into consideration the gap between theory and practice, we provide an explanation of revised ResNets, emphasizing mild assumptions to shed light on the underlying phenomena;

- Finally, we illustrate how applying ideas from dynamical systems theory can enhance the interpretability of ResNets.

## 2 Qualitative properties of Dynamical Systems for ResNets

### 2.1 Preliminary

We construct the preliminary inspired by E (2017); Li et al. (2022). ResNets can be effectively described through the framework of a function $F : \mathbb{R}^n \to \mathbb{R}^m$. We denote this function as $F = g \circ \varphi$, where $\varphi = \varphi_{T-1} \circ \cdots \circ \varphi_0$, and each $\varphi_i : \mathbb{R}^n \to \mathbb{R}^n$ assumes the form:

$$\varphi_i(x) = x + V_i \sigma(W_i x + b_i),$$

where $\sigma : \mathbb{R} \to \mathbb{R}$ represents the (component-wise) activation function, while $W_i \in \mathbb{R}^{p_i \times n}$, $b_i \in \mathbb{R}^{p_i}$, and $V_i \in \mathbb{R}^{n \times p_i}$. The function $g : \mathbb{R}^n \to \mathbb{R}^m$ corresponds to the output layer, often taking the form of a fully connected layer or a softmax layer.

The findings presented in the abovementioned papers highlight the relationship between the output $\varphi(x_0)$ and $x_T$, where the iterative equation $x_{t+1} = x_t + \varphi_t(x_t) = x_t + \varphi_{\theta_t}(x_t)$ encapsulates the weights at time $t$, denoted as $\theta_t = (V_t, W_t, b_t)$. This iterative process can be understood as a discrete-time counterpart to the ordinary differential equation (ODE) represented by:

$$\dot{x}(t) = \varphi_{\theta(t)}(x(t)), \quad x(0) = x_0.$$

The authors of Li et al. (2022) presented the hypothesis space $\mathcal{H}_{ode}$ with respect to $\mathcal{F}$ as follows:

$$\mathcal{H}_{ode}(\mathcal{F}) = \bigcup_{T>0} \mathcal{H}_{ode}(T) \quad \text{where} \quad \mathcal{H}_{ode}(T) = \{g \circ \varphi_T(\cdot; \theta) : g \in \mathcal{G}, \theta \in \Theta \subset \mathbb{R}^q\},$$

Here, $\varphi_T(\cdot; \theta)$ symbolizes the flow-map at time $T$, an essential component associated with the ODE. In practical terms, $\varphi_T(x_0; \theta)$ embodies $x(T)$ where $x : [0, T] \to \mathbb{R}^n$ represents the solution to the aforementioned ODE. The set $\Theta$ encapsulates the parameters of $\varphi_\theta = \varphi_{\theta_t}$ as they manifest in ODE. They termed the set $\mathcal{F} = \{\varphi_\theta : \theta \in \Theta\}$ as the control family, while $\mathcal{G}$ as the terminal family.

While Li et al. (2022) successfully demonstrates the adeptness of approximating any function $g \circ \varphi_T(\cdot; \theta)$ through ODE and ResNets, establishing a direct research connection between Differentiable Dynamical Systems and ResNets is yet to be established. This gap between theory and practice can be attributed to the need for a deeper level of mathematical rigor for a more comprehensive analysis. In this context, we attempted to explore compositions of functions within the domain of $\mathcal{H}$, gaining some insights into their behavior at each processing block, even if it diverges from the conventional ResNet structure. In particular, according to the theory of discrete dynamical systems from a topological point of view, we attempted to examine the dynamics of ResNets. This analysis involves the family of flow and terminal maps, as follows:

$$\mathcal{H} = \mathcal{G} \circ \Phi \quad \text{where} \quad \Phi = \bigcup_{T>0} \{\varphi_T(\cdot; \theta) : \theta \in \Theta\}$$

The compositions of functions serve to portray ResNets as discrete-time variants of ODEs, describing the classification process after defining a control family. (In the manuscript, we assumed $\Phi = \{\varphi\}$ and $\mathcal{G} = \{g\}$ for simplicity.)

### 2.2 Expansiveness

In the realm of dynamical systems, expansivity reveals one of the chaotic phenomena, characterized by an absence of predictability. While it is customary to describe a phenomenon as *chaotic* when it cannot be predicted or explained, various deep learning phenomena exhibit chaotic behavior, where slight changes in inputs can lead to vastly different outputs, signifying a high degree of instability. The concept of expansivity in dynamical systems was first introduced by Utz (1950) under the name *unstable*. However, explaining expansivity in the context of deep neural networks used in computer vision tasks has been challenging. Within the framework of a hypothesis space $\mathcal{H}$, we propose the definition of $g$-expansive, signifying the existence of a residual block where the difference of probability of $\epsilon$ between two images of different classes. Let $X, Y$ be metric spaces and continuous maps $\varphi : X \to X$ and $g : X \to Y$ in what follows.

**Definition 1** We will define that $\varphi$ is *g-expansive* if there exists $\epsilon > 0$ (called *g-expansive constant*) such that whenever $g(x) \neq g(x')$ for some $x, x' \in X$ there exists $n \geq 0$ such that $d(g(\varphi^n(x)), g(\varphi^n(x'))) \geq \epsilon$.

We emphasize that the expansive constant $\epsilon$, represents the minimum distance between the probabilities of pairs of images as they pass through residual blocks and is dependent on the chosen metric. This definition is a new concept in the literature of dynamical systems, and the closest work to ours is in Achigar et al. (2018).

## 2.3 SHADOWING PROPERTY

The original meaning of the *shadowing property* implies that approximated trajectories may be followed by true ones as close as we want. This concept, initially introduced from a topological perspective by Bowen (1975), is fundamental to the qualitative theory of dynamical systems. In this manuscript, we introduce the *g-shadowing property*, which means that there exist images that are not classified as different classes even when passing through each of the residual blocks, with sufficiently small local probability errors, as compared to approximated orbits. Let $\mathcal{H}$ be a hypothesis space. Given $\delta > 0$, we say that a sequence $\xi = (x_n)_{n \in \mathbb{N}_0}$ of $X$ is a $\delta$-pseudo-orbit of $\varphi$ for $g$ if $d(g\varphi(x_n), g(x_{n+1})) \leq \delta$ for all $n \in \mathbb{N}_0$. We say that $\xi$ can be $\epsilon$-shadowed of $\varphi$ for $g$ if there is $x \in X$ such that $d(g\varphi^n(x), g(x_n)) \leq \epsilon$ for any $n \geq 0$.

**Definition 2** We say that $\varphi$ *has the g-shadowing property* if for all $\epsilon > 0$, there exists $\delta > 0$ such that for every $\delta$-pseudo orbits of $\varphi$ for $g$ can be $\epsilon$-shadowed for $g$.

To quantify the degree of stability, we introduce the $g$-shadowing constant as follows. Given $\epsilon > 0$ we can consider the set $S(g, \varphi, \epsilon)$ of $\delta \geq 0$ for which every $\delta$-pseudo orbit of $\varphi$ for $g$ can be $\epsilon$-shadowed for $g$. Note that $[0, \delta] \subset S(g, \varphi, \epsilon)$ for $\delta \in S(g, \varphi, \epsilon)$. So, the supremum $\sup S(g, \varphi, \epsilon)$ represents the optimal number under which every pseudo orbit can be $\epsilon$-shadowed. By dividing it by $\epsilon$ and taking limit inferior as $\epsilon \to 0$ we get the following value called *g-shadowing constant*:

$$Sh_g(\varphi) = \liminf_{\epsilon \to 0} \frac{\sup S(g, \varphi, \epsilon)}{\epsilon}.$$

This constant represents the maximum error of pseudo-orbits concerning the minimal error for rigid images.

## 2.4 TOPOLOGICAL STABILITY

Now, we introduce topological stability for ResNets, motivated by the notions of robustness and stability in the deep learning and dynamical systems literature, respectively. In the context of ResNets, robustness refers to the ability of a trained ResNet to generalize and accurately classify test data, even when the distribution of the test data differs from that of the training data. In particular, the robustness of residual blocks is about perturbations of trained ResNets for each residual block. In connection with dynamical systems, we now present *topological g-stability* on $\mathcal{H}$ based on the classical stability theorem by Walters (2006), which establishes that expansiveness and the shadowing property imply the topological stability of a given system. Denoted by $2^X$ the set formed by the subsets of $X$ and $Id_X$ the identity map of $X$. The supremum distance of continuous maps can be denoted by $d_{C^0}(\varphi, \hat{\varphi}) = \sup_{x \in X} d(\varphi(x), \hat{\varphi}(x))$.

**Definition 3** We say that $\varphi$ is *topological g-stable* on $\mathcal{H}$ if for every $\epsilon > 0$, there exists $\delta > 0$ such that for every continuous map $\hat{\varphi}$ with $d_{C^0}(\varphi, \hat{\varphi}) < \delta$, there is a strict compact $g$-valued upper semi-continuous map $H : X \to 2^X$ such that *(i)* $d(H, Id_X) \leq \epsilon$; *(ii)* $\varphi \circ H \subseteq H \circ \hat{\varphi}$.

Similarly, as with the $g$-shadowing constant, we quantify topological $g$-stability as follows: For all $\epsilon \geq 0$, we define the set $T(g, \varphi, \epsilon)$ of $\delta \geq 0$ for which for every continuous map $\hat{\varphi} : X \to X$ with $d(\varphi, \hat{\varphi}) \leq \delta$ there exists a strict compact $g$-valued upper semi-continuous map $H : X \to 2^X$ such that $(i)$ $d(H, id_X) \leq \epsilon$; $(ii)$ $\varphi \circ H \subseteq H \circ \hat{\varphi}$.

Then, we will call the *topological g-stable constant*:

$$T_g(\varphi) = \liminf_{\epsilon \to 0} \frac{\sup T(g, \varphi, \epsilon)}{\epsilon}.$$

The main theorem is to determine the qualitative properties of solutions of trained ResNets that exhibit stability when passing through residual blocks. We will define $Lip(g) = \sup\{K > 0 :$

$d(g(x), g(y)) \le K \cdot d(x, y), \ \forall x, y \in X\}$ ($K$ is called the Lipschitz constant of $g$). We say that $g$ is *bounded-to-bounded* if $g^{-1}(B)$ is bounded for every bounded subset $B \subset Y$.

**Theorem 1** Let $X \subset \mathbb{R}^n$ and $Y \subset \mathbb{R}^m$ be closed subsets of Euclidean spaces. If $\varphi : X \to X$ is continuous and $g$-expansive for some Lipschitz bounded-to-bounded map $g$, then

$$\color{red} Sh_g(\varphi) \le Lip(g) \cdot T_g(\varphi).$$

See Appendix A for the specific definition of topological $g$-stability and the proof of Theorem 1.

## 3 EXPERIMENTS

In this section, we describe how to apply qualitative properties of dynamical systems to numerical experiments. We address several gaps between theory and practice and modify the model based on the formalism of ResNets.

- To provide a theoretical illustration of ResNets, each residual block requires revision to maintain the same dimensionality $n$ (the number of channels and pixels).

- Theoretically, inputs would enter residual blocks immediately, but in practice, there is a single CNN layer before entering residual blocks. Therefore, the first CNN layer was replaced with a patch to reduce the difference in the effects of the CNN layer before entering residual blocks.

- Trained residual blocks with a number of $T > 0$ can be considered with the parameters as the family of continuous maps $\{\varphi_{\theta_T} \circ \cdots \circ \varphi_{\theta_1}\}$. However, for the purpose of theoretical proof, we present the solution of ResNets using only one parameter denoted as $\Phi = \{\varphi\}$. Similarly, we also consider a terminal family $\mathcal{G} = \{g\}$ as one parameter.

**Datasets.** We used the following datasets for evaluation: the MNIST and CIFAR-10 datasets **?**. The MNIST dataset consists of images of digits and includes 50,000 training images and 10,000 testing images across 10 classes, with a resolution of $28 \times 28$ pixels. To facilitate our experiments, we applied zero-padding to each edge of the images, resulting in the use of the MNIST data with a resolution of $32 \times 32$ pixels. Additionally, we expanded the image channels from 1 to 3. The CIFAR-10 dataset, also including the same number of training and testing images as MNIST, has a resolution of $32 \times 32$ pixels and RGB channels. Both the MNIST and CIFAR-10 datasets are publicly available.

**Model details.** In our experiment, We employed ResNets-18 and ResNets-50 models.

The conventional ResNets consist of four compartments, each of which alters the input data's dimensions. For example, in ResNet-50, the blocks from the 4th to the 7th have 512 output channels, while the 1st to the 3rd have 256 channels for output size. However, for our experiment, we needed to maintain uniform data dimensions. To achieve this, we made several modifications to the ResNets architecture shown in Figure 1. Initially, we retained only the first convolutional layer and the first max-pooling layer, removing all downsample and channel expansion functions. We introduced a patching function to address channel expansion, followed by group convolution to compute convolution for each patch. Finally, we eliminated global pooling at the end of the model to minimize information loss. Additionally, we made some adjustments to the ResNet-18 model, utilizing bottleneck blocks, to ensure that these two models are identical, except for the number of blocks.



Figure 1: Revised ResNets.

**Training ResNets.** To establish the theoretical framework, we made revisions to the ResNets models and conducted training on them. Subsequently, we present the fundamental performance of the

proposed ResNets with respect to the loss on the test dataset. Each classification result was calculated using the scikit-learn package. All models were trained for 100 epochs, utilizing the Adam optimizer and cross-entropy loss with default parameters from the PyTorch package. The learning rate varied depending on the dataset, with MNIST and CIFAR-10 datasets trained at $10^{-4}$. In both MNIST and CIFAR-10 experiments, we used a mini-batch size of 400. For both training and testing, we employed two different machines: one equipped with a 3.2GHz processor and 16GB RAM, and the other with 32GB RAM and an NVIDIA Tesla V100 GPU with 32GB of memory.

Table 1: The basic performance of revised ResNets

| Datasets | Models | Precision | Recall | F1-score | Loss |
|----------|--------|-----------|--------|----------|------|
| MNIST | ResNets-18 | 0.9893 | 0.9891 | 0.9891 | 0.0363 |
| MNIST | ResNets-50 | 0.9860 | 0.9855 | 0.9857 | 0.0449 |
| CIFAR-10 | ResNets-18 | 0.7641 | 0.7657 | 0.7644 | 0.8241 |
| CIFAR-10 | ResNets-50 | 0.7428 | 0.7398 | 0.7400 | 0.8347 |

In Table 1, we obtained classification results through simulations that involved combinations of different datasets and ResNet models. However, ResNets' performance on CIFAR-10 under certain conditions was lower than that of the conventional model. The main difference can be attributed to the method of channel expansion. In particular, the conventional residual block conducts convolution on the entire image with different kernels, even after the image is downsampled. As mentioned in the model details, we replaced pooling between blocks with patching, which directed the model to compute different sections of the image as it went deeper. Consequently, we analyze the reason for the low performance of ResNets with the CIFAR-10 dataset. The backgrounds of the images remain unfiltered, and the model's output contains excessive information. One might question whether all residual blocks can effectively extract features for image processing, but our objective is solely to evaluate the trained ResNets.

**The constants of properties of dynamical systems.** To assess various properties, one can calculate the $l^2$ distances between all pairs of images, corresponding to all values passing through residual blocks and its fully connected layer. Table 2 presents the constants associated with the qualitative properties. Trained ResNets with the number of residual blocks $T > 0$ can be considered the parameterized residual blocks as the family of continuous maps with parametrized fully connected layer $g_{\theta_n}$ for $n = 0, \cdots, T$.

Note that the parameters of the function will vary from model to model, and if trained on different datasets, each parameter will also differ. Thus, all constants, including the Lipschitz constant, depend on the dataset and model.

Table 2: The values of constants for various properties related to dynamical systems.

| Datasets | Models | $g$-expansive | $g$-shadowing | Lip(g) | topological $g$-stable |
|----------|--------|---------------|---------------|--------|------------------------|
| MNIST | ResNets-18 | $4.14 \times 10^{-5}$ | 2.3583 | 3.38 | 0.6977 |
| MNIST | ResNets-50 | $8.75 \times 10^{-6}$ | 0.0031 | 1.77 | 0.0017 |
| CIFAR-10 | ResNets-18 | 0.0081 | 0.9867 | 3.51 | 0.2811 |
| CIFAR-10 | ResNets-50 | 0.0048 | 1.0023 | 2.37 | 0.4229 |

As a result of the $g$-expansive constant of ResNet-18 on MNIST, it can be inferred that for every image pair with different classes, there exists at least one residual block with a distance greater than $4.14 \times 10^{-5}$. Equivalently, when the distance between pairs of images passing through all blocks is less than $4.14 \times 10^{-5}$, the images can be considered as belonging to the same class.

Comparing ResNets-18 and ResNets-50, we observe that ResNets-50 exhibits a tendency to classify images from different classes with a shorter distance, likely attributed to passing through more blocks. Additionally, the solution of trained ResNets on CIFAR-10 tends to be generous for classification, resulting in higher $g$-expansive constants compared to those of MNIST.

The primary purpose of the $g$-shadowing property is to establish the connectivity of topological $g$-stability. To determine the $g$-shadowing constant, we calculated all candidate groups that satisfy the $\delta$-pseudo-orbit of $\varphi$ for $g$. For all images, we selected the most similar image after passing through the block once and calculated the corresponding distance. Specifically, we extracted the maximum value of $\delta$ with the minimal value of $\epsilon$ by using all the calculated values to find the best tracing, as follows:

- (MNIST, ResNet-18) 2.3583 with $\epsilon : 0.1025$, $\delta : 0.2417$;
- (MNIST, ResNet-50) 0.0031 with $\epsilon : 0.1565$, $\delta : 0.0004$;
- (CIFAR-10, ResNet-18) 0.9867 with $\epsilon : 0.0933$, $\delta : 0.0920$;
- (CIFAR-10, ResNet-50) 1.0023 with $\epsilon : 0.1219$, $\delta : 0.1222$.

By utilizing these methods and Theorem 1, we aim to determine the topological $g$-stable constant by multiplying it with the Lipschitz constant of $g$. To calculate the Lipschitz constants of a fully connected network $g$, we utilized the *LipSDP* package developed by Fazlyab et al. (2019), along with LipSDP-Neuron (using a 1K split size and SDPT3 solver). Since this package requires more than 2 layers, we placed an identity matrix for weight before the weight of the fully connected layer we trained on the final one of ResNets. Thus, we can determine the lower bound of the topological $g$-stable constant by Theorem 1 after obtaining the upper bound of the Lipschitz constant.

## 4 APPLICATIONS

In this section, the application of the aforementioned concepts is actively demonstrated. Tracking how trained ResNets consistently classify is a focal point. To emphasize, in the process of calculating the $g$-expansive constant, we measure all values of distances passing through all residual blocks and their fully connected layer.

Figure 2 illustrates the results of analyzing pairs of images that exhibited the minimum distance on average of all residual blocks, while Figure 3 presents the existence of pairs of images that had the closest distance at a specific residual block. In both Figures 2 and 3, the $x$-axis represents the corresponding classes in order, and the $y$-axis represents the value of a certain $g$-expansive constant, respectively. A round dot in the picture indicates the presence of an image of the class represented by a certain $g$-expansive constant.

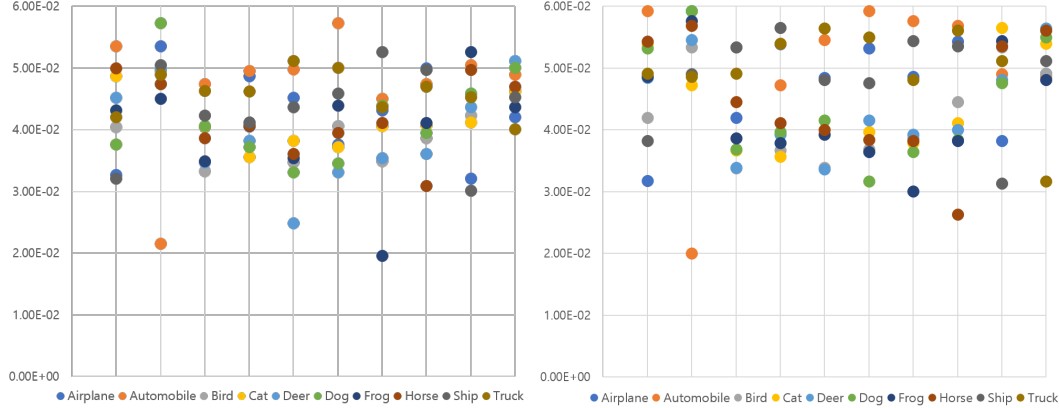

Figure 2: On the first line (class airplane) for ResNets-18 (Left), a gray point (a ship image) indicates the existence of a pair of ship images and airplane images that have a closer distance than when comparing the distances between airplane images. Also, ResNets-18 suspects that its performance may be particularly poor in classifying birds, cats, and trucks than other classes. Compared to ResNet-18, the characteristic of ResNets-50 (Right) is that it can consider representative image pairs that can be well distinguished into the same class while passing through all blocks since the distance of the same class images appears closest.

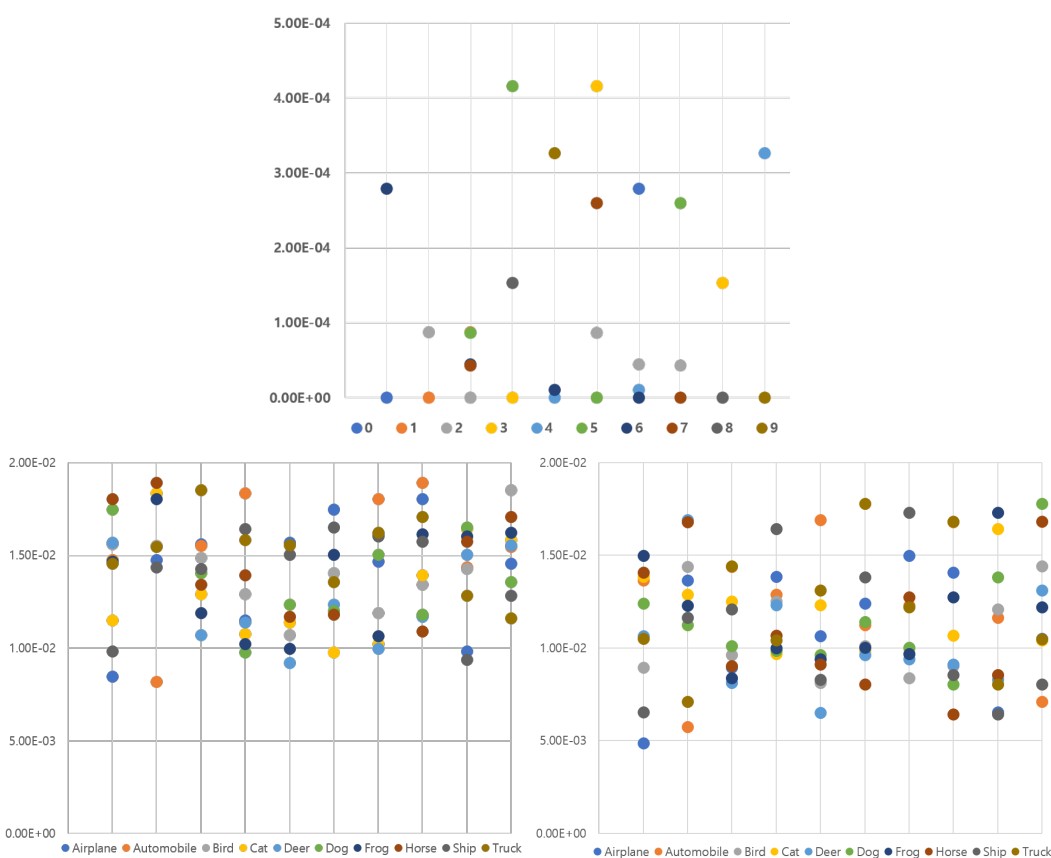

Figure 3: Comparison of $g$-expansive constants for each class. ResNet-50 on MNIST (Top) generally performed well in distinguishing all numbers. However, there were challenging instances, such as distinguishing between the handwritten numbers 4 and 6, and each pair involving the handwritten numbers 2 with 1, 5, 6, and 7, respectively. More details are provided for the handwritten number 2 in Table 3. In both ResNets-18 (Left) and ResNets-50 (Right) on CIFAR-10 (Bottom), there were images in the Bird, Cat, Dog, and Frog classes that appeared less distinguishable than images in other classes. Generally, ResNets-18 tends to struggle with distinguishing between animals. For ResNets-50, a moment when a pair of automobile and truck images were close was observed. In Table 3, we specifically examine the moment with the minimum distance.

Based on the information in Figure 3, details in Table 3 become evident. Considering all distances for images passing through both residual blocks and the fully connected layer allows for the determination of the minimum distance $\epsilon$ corresponding to the $n$-th residual block. The MNIST and CIFAR-10 datasets are obtained using the Pytorch package. Furthermore, each collected image is numbered in order after the dataset is not only used without shuffling but also collected separately into each class which was distributed approximately 1K images per class. Hence, tracking the image pair with the closest distance in one of the residual blocks is pursued. Notably, there were image pairs related to the $g$-expansive constant that exhibited distances slightly smaller than those indicated in Table 3 (specifically, the handwritten numbers 4 and 6 in the case of MNIST). However, we mainly have listed those instances where distinctions were challenging even upon visual examination.

Our perspective on the $g$-expansive constant encompasses complex phenomena in classification tasks in computer vision inspired by the concept of expansivity in the literature of dynamical systems. Its significance lies in our discovery of images in the dataset that can lead to misunderstandings even when viewed by humans. Furthermore, we aim to analyze the sensitivity of classification occurring while passing through residual blocks. Consequently, this offers an opportunity to reexamine aspects often taken for granted, such as the MNIST and CIFAR-10 datasets.

Table 3: Examples showing a close distance for image classification of different class pairs in MNIST (Top) and CIFAR-10 (Bottom).

| Models | ResNet-50 | | ResNet-50 | | ResNet-50 | | ResNet-50 | |
|---|---|---|---|---|---|---|---|---|
| Class | 2 | 1 | 2 | 5 | 2 | 6 | 2 | 7 |
| Images | | | | | | | | |
| Order | $1021_{th}$ | $186_{th}$ | $540_{th}$ | $348_{th}$ | $668_{th}$ | $377_{th}$ | $435_{th}$ | $919_{th}$ |
| n-th block | $1_{th}$ | | $16_{th}$ | | $11_{th}$ | | $14_{th}$ | |
| Distance | $8.67 \times 10^{-5}$ | | $8.63 \times 10^{-5}$ | | $4.43 \times 10^{-5}$ | | $4.24 \times 10^{-5}$ | |
| Models | ResNet-18 | | ResNet-18 | | ResNet-50 | | ResNet-50 | |
| Class | 2 | 4 | 4 | 6 | 0 | 8 | 1 | 9 |
| Images | | | | | | | | |
| Order | $666_{th}$ | $381_{th}$ | $171_{th}$ | $37_{th}$ | $639_{th}$ | $626_{th}$ | $16_{th}$ | $101_{th}$ |
| n-th block | $2_{rd}$ | | $3_{nd}$ | | $6_{th}$ | | $9_{th}$ | |
| Distance | 0.0107 | | 0.0099 | | 0.0065 | | 0.0071 | |

Next, we will show the results of the $g$-shadowing property for assessing ResNet-18 and ResNet-50 on MNIST, CIFAR-10. ResNet-18 discriminated handwritten digit 4, as far as feasible, it traces as a similar thing with the probability errors that may be small as it passes through each of the 8 residual blocks as shown in Table 4. In other words, it can be misunderstood that it has (not intended) stability in distinguishing even though images of other classes appear that are similar to class 4. The results for the handwritten number 4 in Table 4 show that it can be classified as the handwritten number 9 using the somewhat perturbed solution of ResNets.

In the other case in Table 4, there was the trap of a 5th-residual block that traced the image of a ship with misunderstanding to the class of airplane. Moreover, it should be noted that this behavior measures stability constants in Table 2 even in situations involving incorrect solutions of ResNets.

In the MNIST case of Table 5, residual blocks consistently allowed tracing along the same classes while minimizing the probability errors that might occur in each pass through each residual block. Therefore, the $g$-shadowing value, which represents our intended robustness, is 0.0031, as shown in Table 2. As can be seen from the horse image of CIFAR-10 in Table 5, we will need to make assumptions that allow tracing to the same class passing through the residual blocks. It is necessary to adjust the values for $g$-shadowing constants.

Moreover, the $g$-shadowing property means finding slightly perturbed images that approximately follow the solutions of ResNets over time. This provides a bridge between the system's chaotic behavior and the possibility of making short-term predictions within a limited time interval. The stability of ResNets is intended to exhibit shadow properties because small perturbations in initial conditions do not lead to sharp divergence of the classification.

Table 4: Selection of the image that best can be traced via each of residual blocks (ResNet-18).

| Best | MNIST |
|---|---|

| Best | CIFAR-10 |
|---|---|

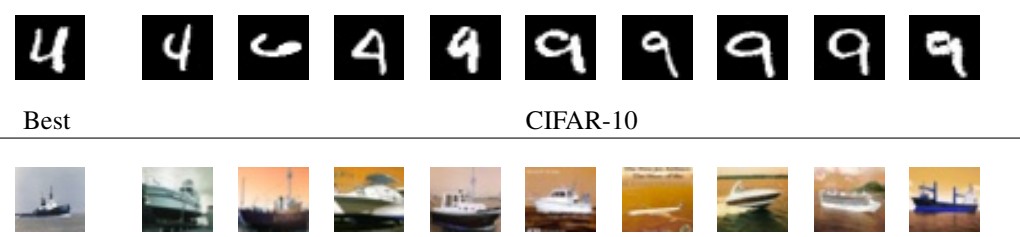

Table 5: Selection of the image that best can be traced via each of residual blocks (ResNet-50).

## 5 CONCLUSION REMARKS

Our primary goal is to advance the foundation of deep learning through the lens of dynamical systems. We initiate our exploration by examining metric spaces derived from high-dimensional Euclidean spaces. Emphasizing the significance of these metric spaces measuring the distances among probabilities, we apply established dynamical systems theory. Despite the ongoing development of research on ordinary/partial differential equations (ODEs/PDEs) and ResNets, our focus lies on bridging dynamical systems itself (induced by ODEs) and ResNets. This common concern in the study of dynamical systems and deep learning involves grappling with the challenge of deriving qualitative (or quantitative) insights.

In this paper, the aim is to provide theoretical insights into dynamical systems supported by numerical experiments. Despite advancements in theoretical understanding, a persistent gap remains between theory and practical implementation in the realm of deep learning. Aligning with the approach presented in this paper, it is imperative to contribute to both theoretical understanding and practical applications to narrow this gap. Additionally, improvement is needed to provide absolute quantitative values. There remains the issue that specific comparisons are only possible when the model and data are different, respectively, or when the model and data are the same, but parameter differences occur due to differences in training.

The main goal is to apply well-established tools of qualitative theory of dynamical systems that has developed over a century, directly to the field of deep learning. To begin with, we evaluate trained ResNets and provide quantitative explainability for stability using the $g$-expansive and $g$-shadowing constants, inspired by fundamental theorem in the literature concerning the qualitative properties of dynamical systems.

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

## A  PROOF OF THEOREM 1.

Let $X \subset \mathbb{R}^n$ and $Y \subset \mathbb{R}^m$ be closed subsets of Euclidean spaces as inputs and outputs for $n, m \in \mathbb{N}$, modeling deep neural networks in general one. Let $\varphi : X \to X$ and $g : X \to Y$ be continuous maps. We define the pseudometric $d_g$ of $X$ by $d_g(x, y) = d(g(x), g(y))$ for all $x, y \in X$ (In fact, $d_g(x, y) = 0$ does not implies $x = y$). We can define

$$E_g(\varphi) = \{e > 0 : \Phi_g(\varphi, e) = \{x\}, \quad \forall x \in X\},$$

where $\Phi_g(\varphi, e) = \{y : d_g(\varphi^i(x), \varphi^i(y)) \le e, \quad \forall i \ge 0\}$

Recall that a sequence $(x_i)$ is a $(\delta, g)$-pseudo orbit if $d_g(\varphi^i(x_i), x_{i+1}) \le \delta$ for all $i \ge 0$. And, we say $(x_i)$ can be $(\epsilon, g)$-shadowed if there is $x \in X$ such that $d_g(\varphi^i(x), x_i) \le \epsilon$, for all $i \ge 0$. Then, we can define $Sh_g(\varphi, \epsilon) = \{\delta \ge 0 : \text{every } (\delta, g)\text{-pseudo orbit of can be } (\epsilon, g)\text{-shadowed}\}$. Thus, we present *g-shadowing constant* $Sh_g(\varphi)$ as dealing with the limit inferior for $\epsilon$.

$$Sh_g(\varphi) = \liminf_{\epsilon \to 0} \frac{\sup Sh_g(\varphi, \epsilon)}{\epsilon}, \text{ where } \sup Sh_g(\varphi, \epsilon) = \sup\{\delta : \delta \in Sh_g(\varphi, \epsilon)\}.$$

Also, we say that a set-valed map $H : X \to 2^X$ is

- strict if $H(x) \ne \emptyset$, for all $x \in X$;
- compact-valued if $H(x)$ is compact for all $x \in X$;
- g-valued if $H(x) \subseteq g^{-1}(g(y))$, $\forall x \in X$, $\forall y \in H(x)$;
- upper semi-continuous if for every $x \in X$ and a neighborhood $U$ of $H(X)$, there is $\delta > 0$ such that for every $H(y) \subseteq U$, and $y \in X$ with $d(x, y) \le \delta$.

In particular, we consider $Top_g(\varphi, \epsilon) = \{\delta \ge 0 : \text{for every continuous map } \hat{\varphi} : X \to X \text{ with } d_{C^0}(\varphi, \hat{\varphi}) \le \delta^1, \text{ there is a strict compact } g\text{-valued upper semi-continuous map } H : X \to 2^X \text{ such that } (i) \ d_{C^0}(H, id_X) \le \epsilon; \quad (ii) \ \varphi \circ H \subseteq H \circ \hat{\varphi}\}$, then we will call the *topological g-stable constant*:

$$T_g(\varphi) = \liminf_{\epsilon \to 0} \frac{\sup Top_g(\varphi, \epsilon)}{\epsilon}, \text{ where } \sup Top_g(\varphi, \epsilon) = \sup\{\delta : \delta \in Top_g(\varphi, \epsilon)\}.$$

**Theorem 1** Let $X \subset \mathbb{R}^n$ and $Y \subset \mathbb{R}^m$ be closed subsets of Euclidean spaces. If $\varphi : X \to X$ is continuous and $g$-expansive for some Lipschitz bounded-to-bounded map $g$, then

$$Sh_g(\varphi) \le Lip(g) \cdot T_g(\varphi).$$

*Proof of Theorem 1.* Pick $e \in E_g(\varphi), e > 0$, and $\delta \in Sh_g(\varphi)$. We assume $Lip(g) > 0$ (i.e. $g$ is not constant) and take any Lipschitz constant $K$ of $g$. We can take $\delta \in Sh_g(\varphi, \frac{e}{2})$.

If $\hat{\varphi} : X \to X$ is continuous with $d_{C^0}(\varphi, \hat{\varphi}) \le \frac{\delta}{K}$ for $x \in X$, then

$$
\begin{aligned}
d_g(\varphi(\hat{\varphi}^i(x)), \hat{\varphi}^{i+1}(x)) &= d(g(\varphi(\hat{\varphi}^i(x))), g(\hat{\varphi}^{i+1}(x))) \\
&\le K \cdot d(\varphi(\hat{\varphi}^i(x)), \hat{\varphi}(\hat{\varphi}^i(x)) \\
&\le K \cdot d_{C^0}(\varphi, \hat{\varphi}) \\
&= K \cdot \frac{\delta}{K} = \delta
\end{aligned}
$$

Then, $(\hat{\varphi}^i(x))$ is $(\delta, g)$-pseudo orbit. Thus, there is $y \in X$ such that

$$d_g(\varphi^i(y), \hat{\varphi}^i(x)) \le \frac{e}{2}, \quad \forall i \ge 0.$$

This proves that the map $H : X \to 2^X$ defined by

$$H(x) = \{y \in X : d_g(\varphi^i(y), \hat{\varphi}^i(x)) \le \frac{e}{2}, \quad \forall i \ge 0\}$$

is strict. Replacing $i = 0$, we have

$$d_g(H, id_X) \le \frac{e}{2}.$$

---

[1]We use the $C^0$ distance in theory, but in practice we do calculations with the $l^2$ distance. This can be adjusted by simply multiplying by a constant as they are equivalent in finite spaces

Now suppose $y, y' \in H(x)$. Then

$$d_g(\varphi^i(y), \hat{\varphi}^i(x)) \leq \frac{e}{2} \quad \forall i \geq 0,$$

$$\text{and } d_g(\varphi^i(y'), \hat{\varphi}^i(x)) \leq \frac{e}{2} \quad \forall i \geq 0.$$

So, $d_g(\varphi^i(y), \hat{\varphi}^i(y')) \leq e$, $\forall i \geq 0$. Thus, we get $y' \in \Phi_g(\varphi, e) \subseteq g^{-1}(g(y))$. This means $H(x) \subseteq g^{-1}(g(y)) \ \forall y \in H(x)$. Thus, $H$ is $g$-valued.

If $y \in H(x)$, then $d_g(\varphi^i(y), \hat{\varphi}^i(x)) \leq \frac{e}{2}$, $\forall i \geq 0$. So,

$$d_g(\varphi^{i+1}(y), \hat{\varphi}^{i+1}(x)) = d_g(\varphi^i(\varphi(y)), \hat{\varphi}^i(\hat{\varphi}(x))) \leq \frac{e}{2} \quad \forall i \geq 0.$$

This implies that $\varphi(y) \in H(\hat{\varphi}(x))$ proving $\varphi(H(x)) \subseteq H(\hat{\varphi}(x))$. Clearly, $H(x)$ is compact for every $x \in X$. Thus, $H$ is compact $g$-valued.

Finally, we have to show $H$ is upper semi-continuous. Suppose by contradiction that it is not so. Then, there are $x \in X$, a neighborhood $U$ of $H(x)$ and sequence $x_n \to x$ such that $H(x_n) \not\subseteq U$ for every $n \in \mathbb{N}$. Then, there is $x'_n \in H(x_n) \setminus U$.

We can see from the definition of $H$ that $d_g(x_n, x'_n) \leq \frac{e}{2}$ namely $d(g(x_n), g(x'_n)) \leq \frac{e}{2}$ for all $n$. Since $x_n \to x$ and $g$ is continuous, $g(x_n) \to g(x)$. Henceforth, there is $R > 0$ such that $d(g(x), g(x_n)) \leq R$ for all $n$. This implies

$$d(g(x), g(x'_n)) \leq d(g(x), g(x_n)) + d(g(x_n), g(x'_n)) \leq R + \frac{e}{2}, \qquad \forall n \in \mathbb{N}.$$

We conclude that

$$g(x'_n) \in B(g(x), R + \frac{e}{2}), \qquad \forall n \in \mathbb{N}.$$

But $H$ is $g$-valued so $H(x_n) \subset g^{-1}(g(x'_n))$ for all $n \in \mathbb{N}$. This together with the above membership yield

$$\bigcup_{n \in \mathbb{N}} H(x_n) \subset g^{-1}(B(g(x), R + \frac{e}{2})).$$

Since $g$ is bounded-to-bounded, we have that $g^{-1}(B(g(x), R + \frac{e}{2}))$ is bounded in $\mathbb{R}^n$. Since $x'_n \in H(x_n)$, we conclude that $x'_n$ is a bounded sequence and so has a convergent subsequence. Up to passing to such a subsequence, we can assume without loss of generality we can assume that $x'_n$ itself converges to some $x' \in X$ since $X$ is closed.

Since $x'_n \notin U$ which is open,
$$x' \notin U.$$
However, the definition of $H$ and $x'_n \in H(x_n)$ imply

$$d_g(\varphi^i(x'_n), \hat{\varphi}^i(x_n)) \leq \frac{e}{2}, \quad \forall i \geq 0, \quad \forall n \in \mathbb{N}.$$

Fixing $i$ and letting $n \to \infty$, we get

$$d_g(\varphi^i(x'), \hat{\varphi}^i(x)) \leq \frac{e}{2}, \quad \forall i \geq 0.$$

Thus, $x' \in H(x) \subseteq U$ implies
$$x' \in U.$$

This is a contradiction. Therefore, $H$ is upper semi-continuous.

All together showed that $\frac{\delta}{K} \in Top_g(\varphi, \frac{e}{2})$. So,

$$Sh_g\left(\varphi, \frac{e}{2}\right) \subseteq K \cdot Top_g\left(\varphi, \frac{e}{2}\right), \quad \forall e \in E_g(\varphi), e \neq 0.$$

Then,

$$\liminf_{e \to 0} \frac{\sup Sh_g(\varphi, \frac{e}{2})}{\frac{e}{2}} \leq K \cdot \liminf_{e \to 0} \frac{\sup Top_g(\varphi, \frac{e}{2})}{\frac{e}{2}}$$

Thus,
$$Sh_g(\varphi) \leq K \cdot T_g(\varphi).$$

Letting $K \to Lip(g)$ above, we get
$$Sh_g(\varphi) \leq Lip(g) \cdot T_g(\varphi)$$

completing the proof.

