# OpenReview forum: "A qualitative theory of dynamical systems for assessing stability in ResNets"
_ICLR.cc/2024/Conference — Submitted to ICLR 2024_

### Official Review · Reviewer_f824 · 2023-10-26

**Soundness:** 3 good
**Presentation:** 2 fair
**Contribution:** 2 fair
**Rating:** 3
**Confidence:** 4

**Summary:**

The authors aim to provide a theory of dynamical systems as applied to ResNets, in order to better understand their stability and robustness properties. They provide experiments using modified versions of the ResNet-18 and ResNet-50 architectures on the MNIST and CIFAR-10 datasets, and they compute constants for different dynamical systems properties applied to these architectures.

**Strengths:**

- The mathematical formulations in the paper seem correct.
- The introduction covers a decent number of recent papers on the intersection between ResNet architecture and dynamical systems theory.

**Weaknesses:**

- The presentation in this paper is poor. There are way too many typos, grammatical errors, and poorly written sentences; often making it difficult to understand what the authors are trying to convey. The poor writing often leads to sentences that come off as grandiose (whether intended or not), such as: _"What sets this study apart from previous research is the solid theoretical framework bridging deep learning and dynamical systems."_
- The related work is incomplete---it is limited to only closely related papers. For example, there's a lot of work connecting the training of deep nets to potential (algorithmic) stability/robustness properties, e.g. https://arxiv.org/pdf/1509.01240.pdf, http://proceedings.mlr.press/v97/du19c/du19c.pdf.
- The Experiments section is weak. The impact of experiments based MNIST and CIFAR-10 needs to be higher every year to justify using such overused datasets, and I don't think the authors demonstrated that impact here.
- The figures in the Applications section (specifically 2 and 3) are poorly constructed and hard to parse, undermining what should be the proof of the ideas laid out in the theory section. These figures do not do a good job of establishing why any of the preceding mathematical formalism is needed and/or what value it brings.
- No code was provided.

**Questions:**

As far as I can tell, the formalism in section 2 applies to a _very_ general class of dynamical systems (essentially $\dot x(t)=\varphi(x,t), y(t) = g(x(t))$); though you keep referencing deep neural networks, images, etc., none of the stability properties are specific to those concepts.

My question is: how sure are you that these are new concepts? For example, take your proposed new concept of "$g$-expansivity": if there is an $\epsilon$ such that
$$
g(x) \neq g(x') \implies d(g(\varphi^n(x)), g(\varphi^n(x')))\geq \epsilon \text{ for some } n\.
$$
I'm a bit skeptical that the above definition is new to dynamical systems.

---

> ### Author Response · Authors · 2023-11-15
>
> Thank you for your valuable comments. Also, we appreciate questions about the $g$-expansive concept. We will try to explain it as concisely as possible for questions.
>
> > The presentation in this paper is poor. The poor writing often leads to sentences that come off as grandiose (whether intended or not), such as: "What sets this study apart from previous research is the solid theoretical framework bridging deep learning and dynamical systems."
>
> We apologize for the deviation from our original intent, which was to emphasize that "this study aims to establish a framework for deep learning and dynamical systems based on the solid theorem".
>
> > The related work is incomplete---it is limited to only closely related papers.
>
> Thank you for introducing related works.
> We reviewed papers roughly with colleagues. Looking at the two papers, we found that they were contributing by researching stability based on qualitative or quantitative characteristics and focusing on optimization issues. In particular, We realized that the paper [1] is philosophically similar to the optimization perspective of our paper. We will update to related papers.
>
>
> > The Experiments section is weak...
>
> What you pointed out is correct. We acknowledge that our applied research lacks impact as we only provided trend analyses in Figures 2 and 3 and showcased examples exclusively from CIFAR-10 in Table 3.
>
> However, we believe that attempting to analyze the complex phenomena occurring while passing through residual blocks will significantly enhance the impact of MNIST and CIFAR-10 datasets for others.
>
> As a result of Table 3, inspired by the $g$-expansive constant, it is evident that when a feature is not captured well, it can be identified as the same image at some point, even they are different classes image. In the case of MNIST in Table 5, a stable image (handwritten digit 0) was identified through the $g$-shadowing property, demonstrating solid classification as it passes through the residual blocks. Except for this case, the rest of the Table 4 and 5 shows vulnerabilities in a sort of adversarial attacks.
>
> To show that $g$-expansive is one of the deserve property to analyze, we will include a detailed analysis of MNIST related to $g$-expansivity (we already have all necessary components).
>
> > The figures in the Applications section (specifically 2 and 3) are poorly constructed and hard to parse.
>
> In Figures 2 and 3, the $x$-axis represents the corresponding class, and the $y$-axis represents the value of the $g$-expansive constant, respectively. A round dot in the picture indicates the presence of an image of the class represented by a sort of $g$-expansive constant.
>
> For example, in the bottom-left case (ResNets-18 on CIFAR-10) in Figure 2, you can see that a gray circle representing the class ``Ship" is observed on the first line (class: Airplane). When comparing the distances between pairs of ship images and airplane images, as well as between pairs of airplane images, it indicates that there are instances where the distances for pairs of ship images and airplane images are closer on average than when comparing the distances between airplane images.
>
> In Figure 3, in the bottom-left case (ResNets-18 on CIFAR-10), the moment when Bird and Deer were close was captured of the existence. Therefore, Table 3 specifically examines blocks and how closely they were distinguished.
>
> In conclusion, Figure 2 was intended to capture the tendency of whether classes were classified as close overall while passing through the blocks for different class images, and Figure 3 was intended to capture which pair of images were close in any one block.
>
> > No code was provided.
>
> If the paper is approved, the code will be released.
>
> > Q. How sure are you that these are new concepts? For example, take your proposed new concept of "$g$-expansivity":
>
> Answer: The concept of expansivity goes back to Utz's paper [2] mentioned in the references.
> There are several generalizations of this concept: Positive expansivity (dealing with not homeomorphism but continuous maps), expansive flows and group actions, expansive random systems, and so on. A common point in those definitions is that they all deal with dynamical systems in a given space $X$ (say), actually these definitions need this constraint. The term $g$-expansive appears in the terminal situation where we have a dynamical system on a space $X$ subject to measurement represented by the terminal map $g$ from $X$ to another space $Y$. This definition is new and the closest some other work in the literature have been to ours is Achigar et al paper [3].
>
> We will include this little history into the revised version.
>
> - References
>
> [1] H. Moritz et al, Train faster, generalize better: Stability of stochastic gradient descent, PMLR, 2016.
>
> [2] W. Utz, Unstable homeomorphisms, Proc. Am. Math. Soc., 1 (1950) 769--774
>
> [3] M. Achigar et al, Observing expansive maps, Journal of London Mathematical Society, 98(2):501-516, 2018.

---

> > ### Comment · Reviewer_f824 · 2023-11-15
> > **response to rebuttal**
> >
> > Thanks for addressing my questions. In particular, I think adding the explanation (you could almost copy that verbatim) as e.g. a figure caption would go a long way.

---

> > > ### Author Response · Authors · 2023-11-17
> > >
> > > Thank you very much for your comments. It was very helpful in the reorganization of the paper. If you have any further questions or require additional clarification, please feel free to reach out at your convenience.

---

### Official Review · Reviewer_RNLk · 2023-11-01

**Soundness:** 2 fair
**Presentation:** 1 poor
**Contribution:** 2 fair
**Rating:** 3
**Confidence:** 4

**Summary:**

The paper proposed a dynamical systems interpretation of ResNets. The novelty here appears to be in the use of Lipschitz coefficients to understand stability of solutions in the context of chaotic behaviour. This is interpreted in metrics of stability and "shadowing".

**Strengths:**

The goal of quantifying the performance of ResNets using metrics is potentially useful for understanding the behaviour and performance of these algorithms.

**Weaknesses:**

The explanation of the given metrics and how to determine them is unclear. The mathematical explanations are also unclear. There is no interpretation of the main Theorem 1 or how it relates to the numerical experiments. In addition, there has been substantial work on dynamical systems interpretations of ResNets and the work does not position itself in this literature. See questions for support for this assessment.

**Questions:**

Specific concerns and questions are listed as follows.

1) I have seen many dynamical systems interpretations of Resnets before. However, very little of this literature is mentioned. The authors should be clearer when differentiating with previous work. A random selection of google scholar entries: "Forward Stability of ResNet and Its Variants","Understanding ResNet from a Discrete Dynamical System Perspective", "Towards Robust ResNet: A Small Step but a Giant Leap"

2) Distance $d$ has been used in Definition 1 as a distance between vectors and the same distance $d$ has been used in Definition 3 as a distance between functions. What distances do authors use?

3) ``In this context, we attempted to explore compositions of functions within the domain of H, gaining some
insights into their behavior at each processing block, even if it diverges from the conventional ResNet
structure.'' -- in what way is this diverging from the ResNet structure?

4) For $g$-expansive constant in Defn 1, the definition says that there exists $n\geq 0$ such that $d(g(\phi^n(x)), g(\phi^n(x'))) \geq \varepsilon$. How do authors find this $n$ to calculate $\varepsilon$ in Table 2?

5) Not sure I understand the definition $Sh_g(\phi,\epsilon)=\sup (\delta : \delta \in Sh_g(\phi,\epsilon))$. Same with $Top_g$.

6) In definition 3, Identity map $Id_X:X\rightarrow X$, but $H:X\rightarrow 2^X$. How do authors compute distance $d(H, Id_X)?$

7) What are the required properties of $X$ in Definitions 1 and 3? Metric space? Banach space?

8) Statement of Theorem 1. Theorem statements should not include a "we".  Also, what is $Y$ in ``g is bounded to bounded''? Also, what does it mean for a hypothesis space to have a Lipschitz constant?

9) I think that $g$-constants and Lipschitz constants do not depend on the dataset, since it is a property of function $g$ and $\phi$. Why do authors separate these constants for different datasets in Table 2?

10) There are no labels in Figure 2 and 3. If $y$-label can be assumed to be a $g$-expansive constant, there is no information about $x$-labels. Also  description and discussion about these figures (and some tables) are missing in the paper. This made it difficult to  interpret Section 4.

11) The authors make constant use of the first person plural in a way which does not encompass the reader. This makes it seem as if the paper were about the authors and not about the result -- e.g. "we tried to look at the dynamics of ResNets," ``In the manuscript, we assumed''

12) The font of labels, legends and titles of all figures should be increased.

Also, there are some typos and minor mistakes.
1) Not sure why ``Dynamical Systems'' is capitalized everywhere
2) Page 3. I don't think that "Note that if $\delta \in Sh_g(\phi, \varepsilon)$ and $\delta < \delta'$, then $\delta' \in Sh_g(\phi, \varepsilon)$." is quite right. It should be $\delta'\leq \delta$.
3) Page 6. typo "Figure 3and Table 3" $->$ "s in Figure 3 and Table 3"
4) Page 9. Typo "initia l condition" $->$ initial condition
5) ``a directly research linked between''
6) Formatting errors in bibliography

---

> ### Author Response · Authors · 2023-11-16
>
> We greatly appreciate your detailed feedback. We almost agree with all your comments. First, we would like to answer the questions you raised. Since most of them are not explained in the paper, we are actively revising the paper to address the issues you raised.
>
> > 1. The authors should be clearer when differentiating with previous work...
>
> Thank you for recommending a paper on ResNets inspired by Dynamical Systems. We were impressed with the study of structure and controlling hyperparameters from the perspectives of stability/robustness, considering the discrete case.
>
> > 2. Definition 3 as a distance between functions. What distances do authors use?
>
> We utilized the supremum distance of continuous maps, as mentioned in the Appendix:
> $$d_{C^0}(\varphi,\hat{\varphi}) = \sup_{x \in X} d(\varphi(x), \hat{\varphi}(x))$$
>
> This term represents the distance between the solutions of trained ResNets and their perturbed counterparts.
>
>
> > 3. In this context, ...even if it diverges from the conventional ResNet structure. -- in what way is this diverging from the ResNet structure?
>
> As demonstrated in studies on ODE and ResNet, the theory "Differential Dynamical Systems" (focus on vector fields) is crucial for understanding the conventional ResNet structure. However, applying this theory directly to ResNets can be challenging. As an alternative approach, our aim is to initially explore topological dynamics to gain insights into their behavior at each processing block. In this sense, we expressed that it diverges from the conventional ResNets structure.
>
> > 4. For $g$-expansive constant, How to find this $n$ to calculate $\epsilon$ in Table 2?
>
> For every test image, we calculate all values across every block and its fully connected layer. So, we can determine $n$ the number of residual blocks, and its corresponding minimum distance, denoted as $\epsilon$.
>
> In other words, we calculate all distance for pairs of images with layers from $$d(g_{\theta_1} \circ \varphi_{\theta_1}(x), g_{\theta_1} \circ \varphi_{\theta_1}(y))$$ to $$d(g_{\theta_T} \circ \varphi_{\theta_T} \circ \cdots \circ \varphi_{\theta_1}(x), g_{\theta_T} \circ \varphi_{\theta_T} \circ \cdots \circ \varphi_{\theta_1}(y))$$
>
> > 5. The definition $Sh_g(\varphi, \epsilon)$.
>
> For $g \in \mathcal{G}$, let $Sh_g : \Phi \times [0, \infty) \to 2^{\mathbb{R^+}}$ be a map defined by $Sh_g(\varphi, \epsilon)$. We can consider $Sh_g(\varphi)$ with the limit inferior for $\epsilon$.
>
> In the case of experiments where $X$ is a finite set, the definition can be replaced by using minimum and maximum values. First, fix $\epsilon>0$, and we can measure the range of $\delta$ in all calculation and select the maximum value of $\delta$ for $\epsilon$. Finally, we can find $\delta$ with the minimum value of $\epsilon$.
>
>
> > 6. In definition 3, How to compute distance $d(H, Id_X)?$
>
> The distance between point and a set can be measured by $$d(H, Id_X) = d(H(x), x) = \inf_{a \in H} d(a,x).$$
>
> This term signifies the degree of perturbation between the solutions of trained ResNets.
>
> However, we did not calculate it, and insights can only be gained when the value of topological stability is known based on Theorem 1, considering the shadowing property and Lipschitz constant.
>
>
> > 7. What are the required properties of $X$ in Definitions 1 and 3? Metric space? Banach space?
>
> Metric spaces.
>
>
> > 8. What is $Y$ in ``g is bounded to bounded''? Also, what does it mean for a hypothesis space to have a Lipschitz constant?
>
> A function from one metric space into another is bounded-to-bounded if the preimage of any bounded subset of the range is a bounded subset of the domain. And the expression "Given the hypothesis space $H$ with Lipschitz constant" means that $g$ is Lipschitz. We will rewrite the statement as "If $g$ is Lipschitz and bounded-to-bounded, then ...".
>
>
> > 9. I think that $g$-constants and Lipschitz constants do not depend on the dataset.
>
> As shown the third assumption for experiments, trained residual blocks with a number of $T>0$ can be considered with the parameted residual blocks as the family of continuous maps with parametrized fully connected layer $g_{\theta_n}$ for $n = 0, \cdots, T$, $$[ (g_{\theta_1} \circ \varphi_{\theta_1}), \cdots, (g_{\theta_T} \circ \varphi_{\theta_T} \circ \cdots \circ \varphi_{\theta_1})].$$
>
> For example, ResNet-18 and ResNet-50 have $T= 8, 16$, respectively.
>
> In conclusion, the parameters of the function will vary from model to model, and if trained on different datasets, each parameter will also differ. Thus, all constants, including the Lipschitz constant, depend on the dataset and model.
>
> However, we present $\varphi$ and $g$ as one parameter for theoretical proof.
>
> > 10. 11. 12..
>
> We appreciate your valuable comments regarding the manuscript's content, font, figures and the typos you pointed out. In particular, we gratitude your observation on our significant mistake, as indicated by $\delta' < \delta$.
>
> We will make the necessary updates soon.

---

### Official Review · Reviewer_YNhW · 2023-11-05

**Soundness:** 4 excellent
**Presentation:** 3 good
**Contribution:** 3 good
**Rating:** 8
**Confidence:** 3

**Summary:**

The paper introduces a novel characterization of the stability of ResNets in terms of the expansiveness property of dynamical systems and compares to the topological stability already used in assessing stability of neural networks. This provide a novel tool to analyze neural networks providing an interesting insight in their behavior. the paper is mostly theoretical, but i does provide an analysis of what the sadowing and expansive parameter can highlight from a ResNet trained to various standard datasets.

**Strengths:**

- Novel approach to characterize stability, founded on a well established body of work in the field of dynamical systems analysis.
- Relatively easy to read ans self-sufficient, even though it has to introduce concepts from a foreign field.
- Interesting result comparing the shadowing property to topological stability.

**Weaknesses:**

- A terse theoretical paper on the stability of ResNets might not be everyone's cup of tea (limited audience)
- The content is very dense, and some parts require to be gone over more than once.
- The experimntal results are merely illustrative, and it is hard to understand whether one might prefer this model of stability to the other in existence

**Questions:**

The main question would be about why this model of stability. There are many others proposed inthe literature, and while yourexperiments illustrates some interesting information about the dataset obtained by this model, it is not clear weather similar information cannot easily  be obtained from other models.

---

> ### Author Response · Authors · 2023-11-15
>
> Thank you for your comments. We appreciate your request for question.
>
> > A terse theoretical paper on the stability of ResNets might not be everyone's cup of tea (limited audience)
>
> >The content is very dense, and some parts require to be gone over more than once.
>
> > The experimntal results are merely illustrative, and it is hard to understand whether one might prefer this model of stability to the other in existence
>
>
>
> We completely agree with the weaknesses mentioned by the reviewer. As you pointed out, the paper should have been written so that we could more easily understand the results. We don't forget your valuable comments during our update of the manuscript. Thank you again.
>
> > Q. The main question would be about why this model of stability. There are many others proposed in the literature, and while your experiments illustrates some interesting information about the dataset obtained by this model, it is not clear weather similar information cannot easily be obtained from other models.
>
> Answer: In previous work, numerous studies have been conducted on ODEs and ResNets. To connect with prior research with the classical theory of Dynamical Systems, we need to establish a direct link between ResNets and Differentiable Dynamical Systems, which involves the study of vector fields. However, this study is yet to be established, and (unexpected) difficulties may arise due to intricacies that require a deeper mathematical rigor and thorough analysis.
>
> In this situation, we tried to venture into exploring compositions of functions within the realm of $ \mathcal{H} $, thus gaining (slightly) insights into their behavior at each processing block, even if it is far from exact ResNet analysis.
>
> In particular, the reason we focused on ResNets induced by ODEs, as mentioned in the global response, is that a successful exploration of Topological Dynamical Systems and ResNets, which maintains dimensional consistency in passing through blocks, can establish a foundation for research in Differential Dynamical Systems (or dynamics on vector fields).

---

### Author Response · Authors · 2023-11-15

We express our sincere gratitude for the reviewer's insightful evaluation of our work's strengths and weaknesses. In response to global feedback, we focus on enhancing the motivation for ResNets and Dynamical Systems within our manuscript.

While other studies highlight the proficiency of ODEs/PDEs for analyzing ResNets, our work stands at topological spaces that can consider a top-down approach to deep learning via dynamical systems. A direct link between the theory of Dynamical Systems and ResNets is not much to be established.

To contribute directly to deep learning, we aim to apply the theoretically well-developed field of the "qualitative theory of dynamical systems", first introduced in 1892 by Poincaré.

A philosophically common goal of Deep Learning and Qualitative Theory of Dynamical Systems could be stated to deal with the structure of the long-time behavior of systems that change over time, even without finding the exact (or approximated) solutions of the system. In particular, both fields have an interest in the chaotic phenomena called "sensitive to initial conditions". The notion of expansivity, introduced in our paper, much stronger concept than sensitive to initial conditions.

In the manuscript, the most crucial point is that the dimensions of the input and output have to be the same in the setting of the theory of Dynamical Systems. So we have only concentrated on on a sort of ResNets models which can be attributed to residual blocks (and it also can be expanded to DenseNets as further study). To achieve this, we proposed revised ResNets and conducted numerical experiments. In the case of CNNs or DNNs with different dimensions per layer, we would need to explore other settings to develop within the same dimensions of inputs and outputs.


(Discrete) ResNets with with T residual blocks can be elegantly formulated as the composition of functions, $f = g \circ \varphi$, where $\varphi = \varphi_{T} \circ \dots \circ \varphi_1$. Our research framework lies within the domain of dynamical systems, a field deeply rooted in the analysis of evolving structures. Early in the exploration of dynamical systems, attention was drawn to systems whose trajectories are expressed as sequences generated by repeated applications of mappings $\phi$. Specifically, sequences $\phi^n(x)$ were studied, where $x$ serves as a starting point and $\phi^n = \phi\circ\overset{n}{\cdots}\circ \phi$. For simplicity, we focus on scenarios where $\varphi_{T} = \cdots = \varphi_1 = \phi$, allowing us to unveil essential dynamics in our paper.


By delving into the dynamics of ResNets through the lens of discrete dynamical systems, we express $\mathcal{H}$ as $\mathcal{G} \circ \Phi$, where $\Phi = \bigcup_{T>0} { \varphi_T ( \cdot ; \theta) : \theta \in \Theta }$. We specifically first explore a theoretical approach, which involves compositions of functions within the realm of $ \mathcal{H} $, and address to a detailed numerical analysis of revised ResNets in which the dimensions of inputs and outputs of all residual blocks are the same.

Our attempt is to depict trained ResNets by introducing the concept of qualitative properties such as expansiveness, shadowing properties, and stability. We aim to provide numerical stability values using Theorem 1 inspired by one of classical Dynamical Systems theory (Walter's stability theorem) as well as analyze the expansive and shadowing constant.

Since a direct method for obtaining a stability value remains elusive, we just have a lower bound of the topological $g$-stability constant by using Theorem 1 based on the $g$-shadowing and Lipschitz constants.


In summarizing the representations of constants, a large $g$-expansive constant implies a greater distance between image classes at specific blocks, while a larger shadowing constant indicates greater stability. So we tried to show Tables and Figures which can assess trained ResNets and reveal insights into ResNet as capturing nuances in image classification. (But, issues about Figures and Tables have to be enhanced as reviewer's pointed out)

We appreciate by expressing gratitude for valuable feedback. We soon will mention typos and questions, and assure active revisions to enhance the paper's quality.

---

### Author Response · Authors · 2023-11-17

We are pleased to inform you that we have uploaded a revised version of our paper, incorporating valuable feedback from reviewers YNhW, RNLk, and f824. In this updated version, the writing associated with the common comments for reviewer's feedback has purple-coded; Also, the writing for YNhW in cyan, RNLk in red, and f824 in blue.

The major revisions made include:

- Updates to the related papers.
- Clarification of Theorem statements.
- Clear expression of numerical results, accompanied by corresponding figures.


Our aim in crafting this revision was to enhance accessibility, ensuring that reviewer (also, a broader audience) can comprehend the intend of paper.

We extend our sincere gratitude to all the reviewers for dedicating their time and expertise to the evaluation of our paper. All reviewer's comments have been invaluable.

---

### Meta-Review · Area_Chair_YZqG · 2023-12-12

**Metareview:**

The paper attempts to interpret Residual Networks from the lens of Qualitative Dynamical Systems theory. It introduces interesting concepts, however, reviewers are critical in that the paper lacks clarity and "maturity". Importantly, the positioning of the work next to previous works that study ResNets and stability, also considering Dynamical Systems, is lacking. Experiments are also on the thinner side, although I believe that provided good and clear theoretical substance, empirical validation can be less emphasized. The authors acknowledge all these points, and while substantial updates were added to the manuscript, the changes are too many to be directly accepted. Another round of reviewing is required.

**Justification For Why Not Higher Score:**

Lack of clarity, lack of structure, insufficient positioning wrt related work.

**Justification For Why Not Lower Score:**

See above.

---

### Decision · Program_Chairs · 2024-01-16

Reject